# Genetic Subtype-Phenotype Analysis of Growth Hormone Treatment on Psychiatric Behavior in Prader-Willi Syndrome

**DOI:** 10.3390/genes11111250

**Published:** 2020-10-23

**Authors:** Andrea S. Montes, Kathryn E. Osann, June Anne Gold, Roy N. Tamura, Daniel J. Driscoll, Merlin G. Butler, Virginia E. Kimonis

**Affiliations:** 1Division of Genetics and Genomics Medicine, Department of Pediatrics, University of California, Irvine, CA 92868, USA; andrea.sm021@gmail.com (A.S.M.); goldj@uci.edu (J.A.G.); 2Department of Medicine, University of California, Irvine, CA 92868, USA; kosann@hs.uci.edu; 3Health Informatics Institute, University of South Florida, Tampa, FL 33620, USA; Roy.Tamura@epi.usf.edu; 4Department of Pediatrics, University of Florida, Gainesville, FL 32610, USA; driscdj@peds.ufl.edu; 5Departments of Psychiatry, Behavioral Sciences, and Pediatrics, University of Kansas Medical Center, Kansas City, KS 66160, USA; mbutler4@kumc.edu

**Keywords:** Prader-Willi syndrome (PWS), PWS molecular classes, PWS genetic subtype–phenotype correlations, natural history, psychiatric behavioral phenotype, growth hormone treatment

## Abstract

Prader-Willi syndrome (PWS) is a complex multisystemic condition caused by a lack of paternal expression of imprinted genes from the 15q11.2–q13 region. Limited literature exists on the association between molecular classes, growth hormone use, and the prevalence of psychiatric phenotypes in PWS. In this study, we analyzed nine psychiatric phenotypes (depressed mood, anxiety, skin picking, nail picking, compulsive counting, compulsive ordering, plays with strings, visual hallucinations, and delusions) recognized in PWS and investigated associations with growth hormone treatment (GHT), deletions (DEL) and uniparental disomy (UPD) in a cohort of 172 individuals with PWS who met the criteria for analysis. Associations were explored using Pearson chi-square tests and univariable and multivariable logistic regression analyses to control for confounding exposures. This observational study of the largest dataset of patients with PWS to date suggested the following genetic subtype and phenotype correlations in psychiatric behaviors: (1) skin picking was more frequent in those with DEL vs. UPD; (2) anxiety was more common in those with UPD vs. DEL; and (3) an increased frequency of anxiety was noted in the UPD group treated with GHT compared to the DEL group. No other significant associations were found between the genetic subtype or GHT including for depressed mood, nail picking, compulsive counting, compulsive ordering, playing with strings, and visual hallucinations. Further studies will be required before any conclusions can be reached.

## 1. Introduction

Prader-Willi syndrome (PWS) is a multisystemic neurogenetic disorder that affects approximately 1/15,000 live births. PWS is found across all races and affects both genders equally [1,2]. Clinical manifestations change dramatically with age. Infants present with severe hypotonia, feeding problems, poor weight gain, and overall failure to thrive. As the individual enters early childhood, they develop hyperphagia with aggressive and obsessive food-seeking behaviors including hoarding and stealing food and eating non-food items or food from the floor or garbage. These behaviors typically lead to morbid obesity and the associated complications of diabetes, obstructive sleep apnea, and right-sided heart failure if the caloric intake is not externally controlled. Individuals with PWS develop hypothalamic dysfunction, which often leads to endocrinopathies including growth hormone deficiency, hypogonadism, and hypothyroidism [3,4]. PWS also has a unique cognitive and behavioral profile which typically presents with delayed motor and language skills, learning disabilities, and an average intelligence quotient (IQ) of 65 [4]. Severe behavioral problems are common and often present as stubbornness, defiance, easy frustrations, and quickness to anger. They often have mood disorders, a striking inability to control their emotions, obsessive tendencies, autistic traits, and are at high risk of developing psychoses in late adolescence or early adulthood, particularly those with uniparental disomy (UPD) [4,5,6].

Prader-Willi syndrome is a genetically heterogeneous disorder caused by three main molecular mechanisms by which the loss of the paternally expressed genes in the 15q11.2–q13 region occurs, generally by paternal interstitial deletions followed by maternal uniparental disomy 15 and imprinting center defects (ICD) [7,8,9,10]. Depending on the molecular mechanism causing the disorder, significant differences in the clinical presentation, primarily related to the behavioral and psychiatric phenotype, may occur [4,10,11,12,13,14]. Those with deletions (DEL) are more likely to have severe behavioral problems, such as self-injury, food-stealing, and compulsive behaviors as well as speech articulation deficits, yet they often have a particular strength with visual-perceptual skills and jigsaw puzzles [4]. Individuals with UPD tend to have significantly higher verbal IQ scores and a higher likelihood to develop psychotic disorders than those with DEL [4,15].

All individuals with Prader-Willi syndrome are at increased risk for psychiatric comorbidities. Lifetime risk for psychotic illness is reported to be up to 60% in those with UPD and 20% in those with deletions, while the general population risk is less than 3.5% [3]. Common psychiatric disorders seen in those with PWS include affective disorders, compulsions, autistic disorders, and psychoses [4]. The mechanisms for psychiatric disturbances in PWS are not well understood; however, growth hormone and insulin-like growth factor (IGF-I) which are important hormones or peptides involving brain and axonal growth with myelination, are deficient in 40–100% of individuals with PWS [16]. Treating these deficiencies with growth hormone therapy is thought to strengthen neuronal signaling, long-term potentiation, and plasticity in hippocampal and other brain regions, thus improving brain growth and resulting in improved cognition [3].

There is no cure for the Prader-Willi syndrome, so treatment is based on the individual’s symptoms. Recombinant growth hormone therapy (GHT) for PWS was approved in the United States in 2000 and has since been widely recognized as a beneficial treatment for the multiple co-morbidities associated with the syndrome [17,18,19,20]. GHT improves linear growth, body composition and lean muscle mass, metabolism and energy expenditure, bone mineral density, and cardiovascular health across all ages of PWS patients [4,16,18]. In addition to the physical improvements, GHT may improve behavior and cognition. Significant improvements in motor development as well as other markers of development, such as language and cognitive ability, have been observed in infants and toddlers treated with growth hormone compared to those who were untreated [17,18,19,20,21].

This study explored the association between PWS genetic subtypes, growth hormone use, and the prevalence of nine psychiatric behaviors using data from a large multi-site cohort.

## 2. Materials and Methods

An 8-year longitudinal observational study was conducted through the Rare Disease Clinical Research Network’s (RDCRN) Natural History PWS and Morbid Obesity Clinical Protocol. This study was reviewed and approved by the Institutional Review Board of the participating sites prior to enrollment (i.e., University of California Office of Research, Irvine (HS# 2007-5605)). Individuals with PWS (*n* = 355) were recruited by experts in this disorder at four research sites: University of Florida (lead site), University of Kansas Medical Center, University of California at Irvine, and Vanderbilt University. Informed consent was obtained from all individuals or their parents or legal representatives. Funding for this study was provided through the Rare Disease Clinical Research Network (RDCRN) by the National Institutes of Health (NIH)/National Institute of Child Health and Human Development (NICHD) [22].

Data on clinical, cognitive, behavioral, PWS genetic subtypes, physical, and body composition measures were collected over multiple visits. The same questionnaires were used at each of the four clinical sites to collect information on screening eligibility, demographics and diagnosis, medication history and concomitant medications, as well as behavior. Data were collected on each participant’s current and past psychiatric behaviors using yes/no responses which represented parent/guardian assessments and were not necessarily diagnosed by mental health professionals.

Participants were enrolled in this study from 7 September 2006 to 31 July 2014. They were invited to return every year until the age of 3 years and biennially if they were over the age of 3 years. Fifty percent of participants came for one visit. Data were entered and stored at the Data Management Coordinating Center (DMCC) at the University of South Florida in Tampa, Florida. The DMCC provided electronic forms for data entry and performed data retrieval and statistical analyses [22].

This study used the largest dataset on patients with PWS to date to explore the association between growth hormone use and the prevalence of nine psychiatric behaviors (depressed mood, anxiety, skin picking, nail picking, compulsive counting, compulsive ordering, plays with strings, visual hallucinations, and delusions) seen in PWS (Table 1). A comparison between growth hormone users and non-growth hormone users was performed and potential dosage effects were assessed by analyzing ages of growth hormone treatment (GHT) initiation and the duration of GHT. The effects of GHT use on PWS genetic subtypes were also investigated. Exclusion criteria included participants under the age of 8 years as the onset of psychiatric disorders is typically in late childhood or adolescence, as well as those who began GHT after their first visit [23]. In this cohort, 172 participants met the criteria for this study.

Study participants were divided into two cohorts, DEL vs. UPD, as well as those who used GHT at any point in their lives and those who had not, and were analyzed with respect to demographics, medications, and the presence of psychiatric phenotypes at their initial visit using descriptive statistics. Although information was also collected on whether the participant had ever experienced psychiatric behaviors, the data were often incomplete so they were not included in the analysis. Changes in psychiatric phenotype at subsequent visits were also not analyzed as half of the participants came to only one visit. Categorical variables (use of growth hormone, psychiatric medication or sex hormone and PWS genetic subtypes) were described with frequencies and percentages, and continuous variables (age at first visit, age at GHT initiation, and GHT duration) were described using the mean and standard deviation. Associations between the use of GHT and psychiatric phenotypes were explored using Pearson chi-square tests. To further explore associations between GHT use, other exposures, and the risk of psychiatric outcome, univariable and multivariable logistic regression analyses were employed. Multivariable logistic regression was used to control for other independent risk factors and possible confounding exposures when a univariable analysis suggested associations between GHT and psychiatric outcomes with a significance level of *p ≤* 0.05. The following covariates were tested: (1) age at visit 1, (2) age at GHT initiation, (3) psychiatric medication use, (4) PWS genetic subtype, (5) sex of participant, (6) sex hormone use, and (7) GHT duration. Age at GHT initiation was strongly correlated with age at visit 1 (r = 0.837) making it uninformative as an independent contributor to outcome risk; therefore, it was dropped from the multivariable analyses. Sex hormone use and the sex of the participant were also not shown to be associated with any of the outcomes by univariable analyses, so these variables were dropped from multivariable analyses. To determine whether the growth hormone had a different effect on outcome risks for different PWS genetic subtypes, an interaction variable was added to the multivariable model between the GHT and PWS genetic subtypes (DEL vs. UPD). The following interaction variables were created: GH*Del (DEL = 1 and UPD = 0) for phenotypes which showed a positive association with the deletion subtype and GH*UPD (UPD = 1 and DEL = 0) for phenotypes which showed a positive association with the UPD subtype. Statistical analyses were performed using SPSS Statistics Software (IBM SPSS Statistics for Windows, Version 21.0 (IBM Corp., Armonk, NY, USA). The data supporting the findings in this paper are available upon request from the corresponding author, Dr. Virginia Kimonis.

## 3. Results

### Variables

Among our cohort of 172 participants with PWS that met our inclusion criteria, 107 (62%) had DEL, 57 (33%) had UPD, and 8 (5%) had ICD. Of those with DEL or UPD (*n* = 164), 116 (71%) were on GHT (73% vs. 67%, respectively; *p* = 0.40). Psychiatric phenotype frequencies for those with DEL vs. UPD were as follows: depressed mood (30% vs. 30%; *p* = 0.98), anxiety (57% vs. 74%; *p* = 0.04), skin picking (82% vs. 63%; *p* = 0.008), nail picking (49% vs. 39%; *p* = 0.30), compulsive counting (19% vs. 16%; *p* = 0.69), compulsive ordering (43% vs. 40%; *p* = 0.74), playing with strings (20% vs. 18%; *p* = 0.66), visual hallucinations (2% vs. 6%; *p* = 0.21), and delusions (5% vs. 7%; *p* = 0.52) (Table 2). There were no significant differences in age between the DEL and UPD subgroups (*p* = 0.09) (Table 3).

After adjusting for the effects of confounding variables, GHT was significantly associated with increased presence of anxiety (OR = 2.7, 95% CI: 1.006–7.426, *p* = 0.05) and delusions (OR = 14.0, 95% CI: 1.262–155.638, *p* = 0.03) (see Table 4). As the number of participants with delusions was low (*n* = 9), the model was run without the additional covariable of psychiatric medication use, and GHT use remained a significant association (*p* = 0.04). PWS individuals with UPD had a higher presence of anxiety than those with DEL (OR = 7.4, 95% 1.760–31.309, *p* = 0.006) and there was a significant interaction between the GHT use and genotype (*p* = 0.04). GHT use was associated with a 3.25-fold increased presence of anxiety in those with UPD vs. a 2.73-fold increased presence in those with DEL.

Age at GHT initiation was intended to be a measure of dosage; however, it was too strongly correlated with age (r = 0.837, *p* < 0.001) to clearly interpret. Individuals who used GHT were significantly younger at visit 1 than those who did not use GHT: 5 (9%) no GHT vs. 50 (91%) GHT were <13 y., 5 (14%) no GHT vs. 30 (86%) GHT were 14–18 y., 15 (35%) no GHT vs. 28 (65%) GHT were 19–26 y., and 27 (69%) no GHT vs. 12 (31%) GHT were >27 y. (*p* < 0.001) (Table 5). Thus, as age at first visit increased, so did the age of GHT initiation. Among participants with ages 8–13 years at visit 1, 57% started GHT by the age of 2 years and 78% started GHT by the age of 6 years. For those aged 19 years or older, none started GHT by the age of 2 years and only 4% started GHT by the age of 6 years.

GHT duration was also investigated in order to measure dosage. Of the 113 individuals in the GHT cohort who provided data on their duration of treatment, 32 (25%) were on GHT for 0.1–3 years, 31 (24%) for 4–9 years, 35 (27%) for 10–12 years, and 15 (24%) for 13–19 years. As anxiety and delusions had statistically significant associations with GHT use, GHT duration was expected to provide additional supporting evidence. No significant associations, however, were found to support this hypothesis (anxiety: *p* = 0.06; delusions: *p* = 0.54) (see Figure 1, Figure 2, Figure 3 and Figure 4).

## 4. Discussion

This study adds to the existing literature regarding PWS genetic subtype–phenotype associations in Prader-Willi syndrome. According to this dataset, UPD is significantly associated with a higher risk for anxiety and deletions are significantly associated with a higher risk of skin picking. These findings are supported by the literature, which reports that those with UPD have greater vulnerability for developing psychoses [24], and those with deletions have higher rates of developing compulsions and self-injury [10,11,14,25].

Based on reports that growth hormone treatment may improve cognition and behavior in individuals with PWS, we speculated that growth hormone treatment would also contribute to a decreased risk of psychiatric behaviors. This hypothesis, however, was not supported by the data in our study. After adjusting for confounding variables, anxiety and delusions were outcomes that had a significant association with growth hormone treatment use, and the data suggested that GHT use was associated with a 2.7 times increased association with anxiety (CI: 1.0–7.4; *p* = 0.05) and a 14.0 times increased association with delusions (CI: 1.3–155.6; *p* = 0.03). These findings were unexpected as there are no documented associations between psychiatric behaviors and GHT. While these findings may be true, they may also be due to chance or other confounding variables that were not captured by the study. One possible explanation is that those receiving growth hormone treatment may also be receiving superior medical care in which psychiatric symptoms are more likely to be detected. It is also plausible that younger individuals experience more anxiety-inducing situations as they transition through school and different living arrangements, whereas older individuals may have more consistency in their routine. Perhaps the most important limitation is that psychiatric behaviors were not necessarily diagnosed by a mental health professional; instead, the data were provided entirely by a parent/guardian report. Notably, out of 164 participants in the study with DEL or UPD, only nine reported delusions, which explains the wide confidence interval, and along with the modest *p* values for both anxiety and delusions, increases the possibility that these may be chance findings.

It was also hypothesized that growth hormone would have a different effect on the deletion vs. UPD subtypes based on reports of psychiatric differences due to genotype. This hypothesis was supported by the data, which suggests that GHT use has a greater effect on increased risk for anxiety for those with UPD than for those with DEL. These findings have not been previously reported in the literature. While this finding may be true, it may also be due to chance or other confounding variables that were not captured by the study.

In order to support the association between exposure and outcome, we investigated age at GHT initiation and GHT duration for the possible evidence of a consistent dosage effect. Age at GHT initiation was uninformative, however, because it was too strongly correlated to age. This is largely because the GHT was FDA approved for PWS patients in 2000 and there was a lag in GHT being adopted as the standard of treatment. Consequently, individuals had a greater chance of initiating GHT at an earlier age if born after 2000, while individuals born prior did not receive GHT at a younger age as the treatment was not available or approved. Therefore, age at growth hormone initiation was found to be a poor measure of dosage and it was dropped from the analysis. As anxiety and delusions had statistically significant associations with GHT use, GHT duration was expected to provide additional supporting evidence. No significant associations, however, were found to support this hypothesis. Although GHT could be associated with anxiety and delusions without treatment duration effects, the finding of a dosage effect typically supports the association between exposure and outcome. It is possible that dosage truly does have an effect on the outcome, but that duration of treatment is not a good stand-alone measure of dosage. Notably, the type of growth hormone treatment and the dosage of each treatment was not included in the analysis. Another possibility is that GHT duration truly is associated with the outcome, but the sample was not large enough to produce a statistically significant association. The lack of an association with GHT duration, however, suggests that the association of GHT use and an increased risk of anxiety and delusions may be a chance finding.

Psychiatric behaviors are complex and often due to a combination of several genetic and environmental factors [26]. Measuring the effects of one exposure on psychiatric outcome is therefore complicated, as one must attempt to eliminate the effects of all other confounding variables. The strengths of this study include the large longitudinal sample size and the amount of information gathered on each patient using standardized protocols which allowed for the evaluation of several possible confounding variables; however, many additional potential confounders were not included in the data collection and therefore not controlled in the analyses. Potential confounders include race, culture, socioeconomic status (SES), family history, and comorbidities. It is estimated that children and adolescents from families with low SES are up to three times more likely to develop mental health problems than their peers from families with high SES [27]. Access to GHT is also associated with SES due to the expense and not all families have equal access due to variable health care coverage throughout the nation [28]. Additionally, controlling for the co-occurrence of sleep disturbances is also important, as it is common in individuals with Prader-Willi syndrome. Sleep is an important psychophysiological process to promote healthy brain function and mental health, possibly making it a confounding variable [29].

In this study, information on psychiatric diagnosis was gathered by guardian report and data were obtained through medical records rather than through a formal psychiatric evaluation. Psychiatric disorders are highly stigmatized which may lead to the underreporting of symptoms [30]. Stigma regarding mental health disorders has been reported to vary among individuals and families from different races and cultures [31]. Differences in reporting by race and/or culture were not assessed in the analyses in this study, as 85% of the study participants were white; therefore, an important confounding factor for psychiatric behavior reporting may have been missed. A better evaluation of psychiatric disorders in the family history is also recommended, as psychiatric disorders are highly heritable.

## 5. Conclusions

This observational study of the largest dataset of PWS patients to date suggested differences in psychiatric phenotype exist between genetic subtypes, namely those with DEL were more likely to exhibit skin picking while those with UPD were more likely to experience anxiety. The data also suggested the association of GHT and psychiatric phenotype may differ by genetic subtype as an increased frequency of anxiety was found in the UPD group treated with GHT compared to the DEL group. However, in order to better understand the effects of GHT on psychiatric behavior in PWS, the limitations of this study must be addressed. Future analyses should include detailed information on the age of onset of psychiatric symptoms as well as the duration and the frequency of episodes. It would also be important to determine when, and if, symptoms started in relation to growth hormone use. Increasing the sample size to include more adolescent and adult individuals would also be beneficial in order to give the study greater statistical power over a wider age range of individuals with PWS. The authors encourage more controlled studies and prospective investigations in a large PWS cohort similarly treated and assessed to further characterize or validate our preliminary observations of GHT effects on psychiatric behavioral phenotypes in PWS.

## Figures and Tables

**Figure 1 genes-11-01250-f001:**
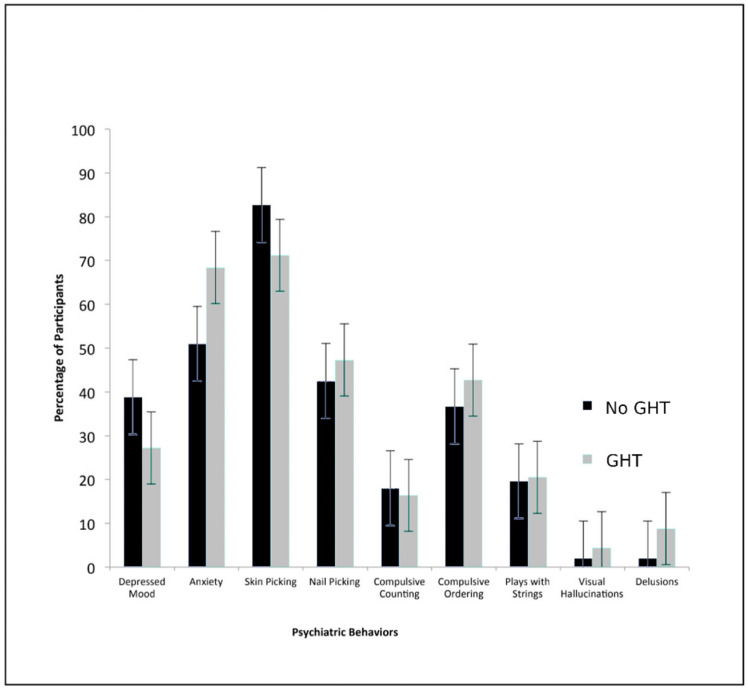
Association between psychiatric behavior and growth hormone treatment use. Percentage of GHT users and non-GHT users who reported the presence of each psychiatric behavior (unadjusted for confounders) at their first visit. There was a significantly higher prevalence of anxiety in GHT users over non-GHT users (*p* = 0.03).

**Figure 2 genes-11-01250-f002:**
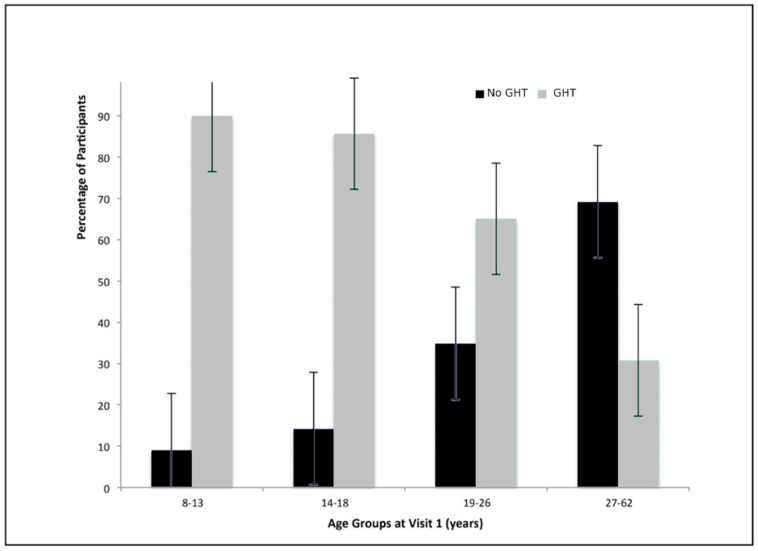
Growth hormone treatment (GHT) use by age group at visit 1. Individuals who used GHT were significantly younger at visit 1 than those who did not use GHT.

**Figure 3 genes-11-01250-f003:**
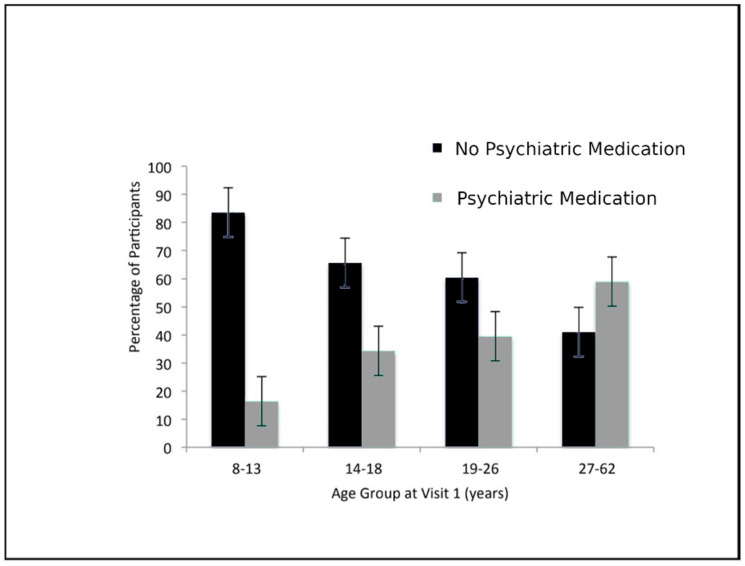
Psychiatric medication use by age group at visit 1. Individuals who used psychiatric medications were significantly older at visit 1 than those who did not use psychiatric medications.

**Figure 4 genes-11-01250-f004:**
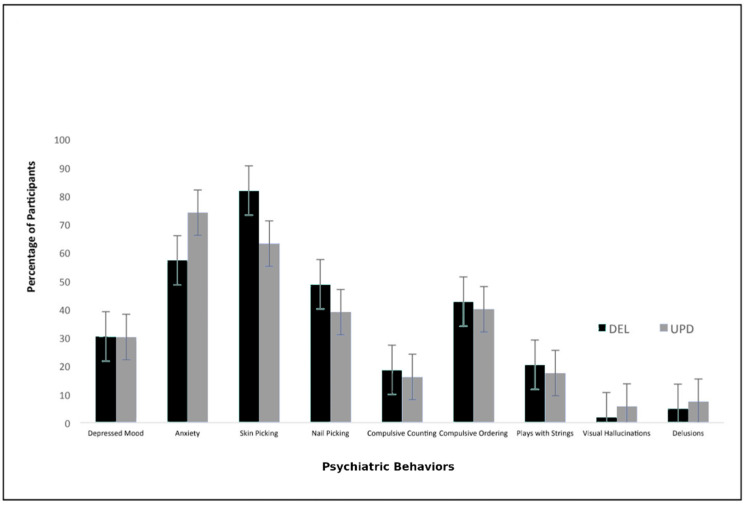
Prader-Willi syndrome (PWS) genetic subtype–phenotype associations (DEL = deletion; UPD = uniparental disomy). Comparison between individuals with DEL and UPD for nine psychiatric phenotypes.

**Table 1 genes-11-01250-t001:** Psychiatric behaviors grouped into three categories.

Depressive Disorders	Compulsions	Psychoses
Depressed mood	Skin picking	Visual hallucinations
Anxiety	Nail picking	Delusions
	Compulsive counting	
	Compulsive ordering	
	Plays with strings	

**Table 2 genes-11-01250-t002:** Presence of psychiatric phenotypes by deletion (DEL) and uniparental disomy (UPD) subtypes.

Phenotype	Presence of Phenotype at Visit 1	DEL		UPD		Totals		Pearson Chi-Square *p*-Value
		*n* *	%	*n* *	%	*n* *	%	
Depressed Mood *n* = 155 *	Yes	31/102	30	16/53	30	47	30	0.98
Anxiety *n* = 157 *	Yes	59/103	57	40/54	74	99	63	0.04
Skin Picking *n* = 162 *	Yes	86/105	82	36/57	63	122	75	0.008
Nail Picking *n* = 127 *	Yes	42/86	49	16/41	39	58	46	0.30
Compulsive Counting *n* = 158 *	Yes	19/102	19	9/56	16	28	18	0.69
Compulsive Ordering *n* = 158 *	Yes	44/103	43	22/55	40	66	42	0.74
Plays with Strings *n* = 160 *	Yes	21/103	20	10/57	18	31	19	0.66
Visual Hallucinations *n* = 156 *	Yes	2/103	2	3/53	6	5	3	0.21
Delusions *n* = 156 *	Yes	5/102	5	4/54	7	9	6	0.52

* Total number of participants varies for each phenotype depending on whether the data were provided or left incomplete on the questionnaire.

**Table 3 genes-11-01250-t003:** Age groups at visit 1 by deletions (DEL) and uniparental disomy (UPD) genotypes.

Age Groups at Visit 1 (Years)	DEL		UPD		Totals		Pearson Chi-Square *p*-Value
	*n* = 107	%	*n* = 57	%	*n* = 164	%	
8–13	29	27	25	44	54	33	0.09
14–18	26	24	7	12	33	20	
19–26	27	25	11	19	38	23	
27–62	25	23	14	25	39	24	

**Table 4 genes-11-01250-t004:** Covariables included in the multivariable model for anxiety and delusions.

Anxiety (*n* = 99)	Delusions (*n* = 9)
GHT use (OR = 2.7, CI: 1.0–7.4; *p* = 0.05)	GHT use (OR = 14.0, CI: 1.3–155.6, *p* = 0.03)
Age at visit 1 (OR = 1.0, 95% CI: 0.9–1.0; *p* = 0.03)	Age at visit 1 (OR = 1.1, 95% CI: 1.0–1.1; *p* = 0.08)
Psychiatric medication use (OR = 3.9, CI: 1.6–9.2; *p* = 0.002)	Psychiatric medication use (OR = 3.4, 95% CI: 0.91–12.8; *p* = 0.07)
UPD genotype (OR = 7.6, CI:1.8–32.1; *p* = 0.006)	
Interaction between GHT use and genotype (OR = 0.2, CI: 0.03–0.9; *p* = 0.04)	

OR = odds ratio; CI = confidence interval.

**Table 5 genes-11-01250-t005:** Descriptive data by growth hormone treatment use.

Descriptive Data		Growth Hormone Treatment Use						
		No		Yes		Total		Pearson Chi-Square *p*-Value
		**n = 52**	**%**	**n = 120**	**%**	**n = 172**	**%**	
Sex	Male	19	37	59	49	78	45	0.13
	Female	33	63	61	51	94	55	
Genetic Subtype	DEL	29	56	78	65	107	62	0.33
	UPD	19	36	38	32	57	33	
	ICD	4	8	4	3	8	5	
Age Group at Visit 1 (Years)	8–13	5	10	50	42	55	32	<0.001
	14–18	5	10	30	25	35	20	
	19–26	15	28	28	23	43	25	
	27–62	27	52	12	10	39	23	

DEL = deletion; UPD = uniparental disomy; ICD = imprinting center defect.

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
