# Peer review of "Genetic Subtype-Phenotype Analysis of Growth Hormone Treatment on Psychiatric Behavior in Prader-Willi Syndrome"

_genes, 2020, doi:10.3390/genes11111250_

Round 1

Reviewer 1 Report

Montes et al describe a large, multicenter natural history study exploring the relationship between several psychiatric phenotypes in Prader-Willi syndrome (PWS) and variables including genetic subtype and growth hormone therapy (GHT) status.  While most psychiatric symptoms do not have clear associations with the variables examined, some (e.g., UPD and increased anxiety) are consistent with other studies, while others (e.g., GHT and increased anxiety and delusions) have not been previously reported and have potentially important implications for clinical care of these patients.

Critique:

GHT has been found in multiple, well controlled prospective studies to positively impact PWS growth, development, and quality of life. Given that the stated conclusions of this study regarding GHT (i.e., that GHT is associated with increased anxiety and delusions in PWS) have the potential to negatively impact GHT prescribing practices and patient access, the limitations of the study should be fully considered when drawing conclusions.  

Major concerns:

There should be more clarity on how the presence of psychiatric behaviors was determined.  For example, how is “anxiety” is defined? Does a positive response include only clinical anxiety conditions, as diagnosed by a mental health professional, or does it represent parent/guardian assessed anxiety, inclusive of mild to moderate anxiety problems?  Was the determination of whether a subject was positive or negative for anxiety based on the study-specific questionnaires (and if so, how were these validated) or on the BASC-2 Parent Rating Scale, or both?  Line 110 is not entirely clear but seems to state BASC-2 was used only for participants <22 years old; if so, what assessment was used for those 23+?

There are several temporal aspects of the study that are incompletely described.  On average, how many visits over how many years were the participants involved? If there were multiple visits, were responses related to the initial assessment only included in the current analysis (as per Figure 1) or were the same questionnaires used at subsequent visits.  In the case of multiple visits and assessments, how were discrepancies in psychiatric symptoms reported at different visits managed? Did the GHT (yes/no) determination relate only to treatment at the time of the visit, or was the question whether the subject had ever used GHT?

If anxiety was not present at the time of evaluation, was the parent/guardian asked whether the subject had ever experienced anxiety, and if so, how was that incorporated into the analysis? 

Even in light of the attempts to control for that potential age effects, age differences in the +/- GHT groups remain a concern.  The authors mention that they control for age at visit 1 as a covariate, but it is unclear whether that is used, especially between GH use and anxiety. As there is such a wide discrepancy between GH use and age (as shown in Table 2), and anxiety is likely highly age dependent, it would be helpful to clarify these relationships with age. Of particular concern here is the low number of subjects not receiving GHT (n=10) in the age range 8-18 years old, where anxiety peaks in typically developing children. It seems plausible that adults who have had a consistent living arrangement for a number of years would have decreased anxiety compared to those subjects in childhood/adolescent years, who are experiencing ongoing transitions in the school setting.

Was there an analysis of the dose of GH, IGF-1 levels, or duration of GHT, to understand more about the relationship of GH exposure and these behaviors? As discussed in section 3.1, age at GHT initiation wasn’t informative, but it’s unclear if dose, total duration of GHT (or % of life on GHT) were associated with the behaviors.

Additional information on how the target psychiatric symptoms change with age and over time in this cohort would be interesting, and presumably could be evaluated from the current data. 

In the 3.1 Variables section, please include N and percentage rather than just percentage to provide a clearer picture. It would be helpful to show the N for the different psychiatric behaviors (both with and without growth hormone). Currently, it is difficult to determine the true differences in psychiatric behaviors by growth hormone status with only Figure 1 to go by.

It’s not clear the actual number of people with delusions in each category, but from Figure 1, it looks like only 1 person without GHT reported delusions, consistent with the wide confidence interval.  The presence of delusions is unexpectedly low given the number of adults with PWS included, and the statement in the introduction that 60% of those with PWS by UPD and 20% of those with PWS by deletion experience psychotic illness.  Please provide additional information regarding the how the occurrence of delusions was assessed (current symptoms only, or current and past? Diagnosed by a medical professional? Controlled by psychiatric medication? Age at diagnosis?)

The conclusions regarding growth hormone use and anxiety and delusions in the discussion section are overstated. Given the extremely wide 95% CIs (which includes 1.0  for anxiety: 95% CI: 1.006-7.426; delusions: 1.262-155.638) and very modest p-values (especially anxiety at 0.049), it is not appropriate to use only the point estimates in the discussion section when discussing GH use and the association with anxiety and delusions.

With small numbers in the subsets, a significant skewing of the population with respect to age vs. GHT, and modest p values for most of the associations, it needs to be made very clear how unstable these estimates are, and to caution any interpretation made from these data.

The abstract and discussion states that anxiety is more common in those with UPD (p=0.038), but Figure 4 legend states that a significant correlation was not found between genetic subtype and anxiety.

Minor concerns:

Sentence structure in lines 68-72 needs correcting; there may be a few missing words.

The abstract should state that there was no association of genetic subtype or GHT with depressed mood, nail picking, compulsive counting, compulsive ordering, playing with strings, and visual hallucinations.

With respect to the BASC-2, was the overall Behavioral Symptoms Index or any additional clinical subscales (e.g., externalizing problems, conduct problems) assessed in relationship to the variables discussed?

Lines 157-159 appear flipped (states that 9.1% of the under 13 year olds received GHT and 90.9% no GHT).    

The statements of non-significant associations of higher risk (lines 204-207) are confusing.  In particular, the increased risk for visual hallucinations and delusions (UPD) is complicated by the very low numbers appear of positive subjects, and the percent of subject with depressed mood appears almost identical (Figure 4) between subtypes, whereas the sentence implies deletion is higher risk.

Reviewer 2 Report

This is an interesting study looking at the association between growth hormone use and psychiatric conditions. However, there are some methodological issues which need to be addressed.

  • Please describe in the methods how often the measures were repeated for individual participants (e.g., annually) and provide information on the length of follow up for individuals participants (e.g., number of years each person was followed).
  • The data were collected longitudinally, but it is not clear from the methods how this was accounted for in analysis. For example, were repeated measures accounted for in logistic regression by modeling the covariance between measures on the same participant?
  • Did reported behaviors change over time for participants, and if so, how was this handled in analysis? For example, in the unadjusted analysis was someone classified as having anxiety if they were classified that way at any one visit?
  • Did growth hormone use status change over time for participants, and if so, how was this handled in analysis?
  • Including psychiatric medication use as a covariate in modeling is statistically problematic. Assuming these medications are only prescribed to those diagnosed with psychiatric illness, adjusting for their use in modeling of psychiatric outcomes does not make sense statistically as it would be in the causal pathway for the outcome, and this impacts the interpretability of the results. Use of these medications seems like it could be an outcome itself.
  • Recommend including a comparison of characteristics between the UPD and deletion groups. If there are chance imbalances between the groups (for example, with respect to age), then the comparisons of psychiatric phenotypes in UPD vs. deletions (in Section 3.1, lines 148-153, and Figure 4) will be more interpretable if they are adjusted for those unbalanced characteristics.
  • Section 3.1, last paragraph: Please state explicitly which variables were included in modeling as covariates.
  • Suggest using the following common formatting conventions for statistical reporting
    • Percentages with denominators of 100 or less should be rounded to whole numbers
    • ORs should be rounded to one decimal place
  • Please include a table showing the number of participants reporting each behavior/psychological condition. Some of the numbers appear to be quite small, and this limitation of the data should be made transparent in the results section. Also, when there is a small number with the outcome (for example, only 9 reporting delusions per the discussion), the number of covariates included in modeling should be limited accordingly.
  • In Table 2, please include the total N in the column headers
  • In Table 2, since there are more individuals in the growth hormone group, it would be more informative to report percentages within the growth hormone and no growth hormone groups (report column percentages rather than row percentages).
  • Figure 1: Since behaviors are known to change with age, it seems important to control for age when evaluating the relationship between growth hormone and behaviors. This is especially true since the growth hormone group was younger at the first visit. Thus, recommend reporting the relationship between growth hormone adjusted for age, rather than unadjusted, so that the results will be more interpretable.
  • Figure 4: The legend states that skin picking was the only statistically significant condition, but this conflicts with the text in section 3.1 that states anxiety was higher in UPD. The legend also refers to “significant correlation” but since Chi-square tests were used, “significant association” would be more appropriate.
  • The discussion should include the limitation that some of the numbers were small with respect to reported outcomes and also subgroups (e.g., numbers with UPD).

Round 2

Reviewer 1 Report

The manuscript is improved, but some additional changes are still needed. The inability to control for age in evaluating the relationship between GHT and behaviors remains troubling and seems a likely basis for differences between GHT and nonGHT groups.   

  1. The abstract states that the investigators analyzed the psychiatric phenotypes over an 8 year period, which implies a longitudinal study, but the data presented for all participants represents current problems from a single visit (visit 1).  The abstract should be modified to clarify that the data are from a single time point. 
  2. Lines 163 and 164 are confusing (GHT had a greater association with increased presence of anxiety for those with UPD than for those with DEL (OR=0.2, 95% CI: 0.03-0.9, p=0.03).)  -  if it was a greater association, the OR would be above 1. Please ensure that the comparison in the text matches the OR.
  3. The sentence in the Discussion with the point values (line 223) should include the confidence intervals for 2.7 and 14 x increased associations.
  4. The discussion (probably 2nd paragraph, before line 228)  should also recognize the age difference in the GH vs. no GH groups, noting that those not receiving GHT were older and used more psychiatric drugs. Finally, the authors may want to acknowledge that prospective, RCT have not detected increased delusions or psychiatric behaviors in those receiving GHT. 

Reviewer 2 Report

Thank you for the changes and clarifications you provided regarding the analysis methods and results. Seeing the information that is added in this version, I have a few more comments.

  1. Since “multivariate” regression refers to analyzing multiple outcome variables in the same model, please refer to the analysis as either “univariable and multivariable” or “unadjusted and adjusted” logistic regression.
  2. Thank you for including the comparison of ages in Deletion and UPD participants in Table 4. To help the reader make comparisons between the genetic subgroups, the percentages should be column percentages (with denominators of the N in the Deletion and UPD groups, respectively) rather than row percentages (with denominators of the N in the age category). Percentages for the Total column are calculated this way already, and the percentages in the Del and UPD columns should be calculated as in the Total column.
  3. In the statement “GHT had a greater association with increased presence of anxiety for those with UPD than for those with DEL (OR=0.2, 95% CI: 0.03-0.9, p=0.03),” it is unclear if the OR is for GH vs no GH among those with UPD, or some other comparison. Please clarify.
  4. In the statement “those with UPD who used GH had a 3.25 fold increased presence of anxiety; whereas those with DEL who used GH had a 2.73 fold increased presence of anxiety,” please instead report ORs and CIs for consistency with the way the other results are reported. Also, typically the way interactions would be presented is to compare one variable within levels of the other variable, for example, presenting ORs for GH vs no GH within the UPD group and within the deletion group (rather than using the group with deletions and no GH as the comparator for both deletions and UPD with GH).
  5. Since there are only 9 individuals with delusions, I strongly suggest only controlling for age and not for psychiatric medication. A rough rule of thumb for logistic regression model is one predictor for every 10 observations in the smaller outcome group – which in this case has n=9. Following that rule of thumb, you would not have room to control for any covariates, however, since GH could be related to delusions strictly because it is also a proxy for age (older patients much less likely to have GH), it seems important to control for age in this case.
  6. Thank you for providing the covariates that were included in modeling. It would be much more informative to include the ORs and CIs rather than the p values (preferably in a table rather than in the text).
  7. Minor comment: since the denominators are provided in Table 3, the “no” category rows could be removed to make the table simpler.
